# Generating Multi-Agent Trajectories using Programmatic Weak Supervision

**Eric Zhan**
Caltech
ezhan@caltech.edu

**Stephan Zheng**[*]
Salesforce
stephan.zheng@salesforce.com

**Yisong Yue**
Caltech
yyue@caltech.edu

**Long Sha & Patrick Lucey**
STATS
{lsha,plucey}@stats.com

## ABSTRACT

We study the problem of training sequential generative models for capturing co-ordinated multi-agent trajectory behavior, such as offensive basketball gameplay. When modeling such settings, it is often beneficial to design hierarchical models that can capture long-term coordination using intermediate variables. Furthermore, these intermediate variables should capture interesting high-level behavioral semantics in an interpretable and manipulatable way. We present a hierarchical framework that can effectively learn such sequential generative models. Our approach is inspired by recent work on leveraging programmatically produced weak labels, which we extend to the spatiotemporal regime. In addition to synthetic settings, we show how to instantiate our framework to effectively model complex interactions between basketball players and generate realistic multi-agent trajectories of basketball gameplay over long time periods. We validate our approach using both quantitative and qualitative evaluations, including a user study comparison conducted with professional sports analysts.[1]

## 1 INTRODUCTION

The ongoing explosion of recorded tracking data is enabling the study of fine-grained behavior in many domains: sports (Miller et al., 2014; Yue et al., 2014; Zheng et al., 2016; Le et al., 2017), video games (Ross et al., 2011), video & motion capture (Suwajanakorn et al., 2017; Taylor et al., 2017; Xue et al., 2016), navigation & driving (Ziebart et al., 2009; Zhang & Cho, 2017; Li et al., 2017), laboratory animal behaviors (Johnson et al., 2016; Eyjolfsdottir et al., 2017), and tele-operated robotics (Abbeel & Ng, 2004; Lin et al., 2006). However, it is an open challenge to develop *sequential generative models* leveraging such data, for instance, to capture the complex behavior of multiple cooperating agents. Figure 1a shows an example of offensive players in basketball moving unpredictably and with multimodal distributions over possible trajectories. Figure 1b depicts a simplified Boids model from (Reynolds, 1987) for modeling animal schooling behavior in which the agents can be friendly or unfriendly. In both cases, agent behavior is *highly coordinated and non-deterministic*, and the space of all multi-agent trajectories is naively exponentially large.

When modeling such sequential data, it is often beneficial to design hierarchical models that can capture long-term coordination using intermediate variables or representations (Li et al., 2015; Zheng et al., 2016). An attractive use-case for these intermediate variables is *to capture interesting high-level behavioral semantics in an interpretable and manipulable way*. For instance, in the basketball setting, intermediate variables can encode long-term strategies and team formations. Conventional approaches to learning interpretable intermediate variables typically focus on learning disentangled latent representations in an unsupervised way (e.g., (Li et al., 2017; Wang et al., 2017)), but it is challenging for such approaches to handle complex sequential settings (Chen et al., 2017).

---

[*]Research done while author was at Caltech.

[1]Code is available at https://github.com/ezhan94/multiagent-programmatic-supervision.

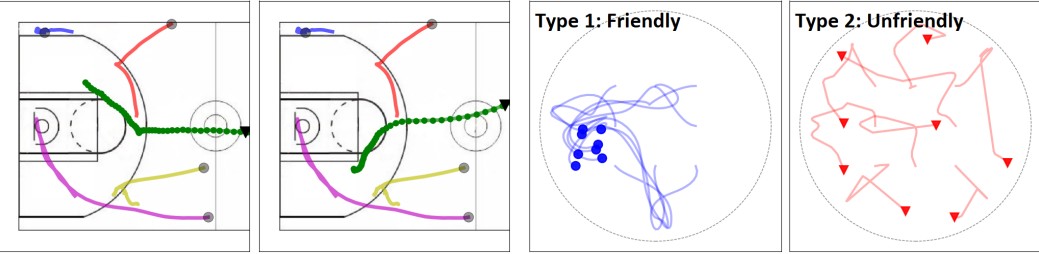

(a) Offensive basketball players have multimodal be-
havior (ball not shown). For instance, the green player
(▼) moves to either the top-left or bottom-left.

(b) Two types of generated behaviors for 8 agents in
Boids model. **Left**: Friendly blue agents group to-
gether. **Right**: Unfriendly red agents stay apart.

Figure 1: Examples of coordinated multimodal multi-agent behavior.

To address this challenge, we present a hierarchical framework that can effectively learn such se-
quential generative models, while using programmatic weak supervision. Our approach uses a label-
ing function to programmatically produce useful weak labels for supervised learning of interpretable
intermediate representations. This approach is inspired by recent work on data programming (Rat-
ner et al., 2016), which uses cheap and noisy labeling functions to significantly speed up learning.
In this work, we extend this approach to the spatiotemporal regime.

Our contributions can be summarized as follows:

- We propose a hierarchical framework for sequential generative modeling. Our approach is
  compatible with many existing deep generative models.
- We show how to programmatically produce weak labels of macro-intents to train the in-
  termediate representation in a supervised fashion. Our approach is easy to implement and
  results in highly interpretable intermediate variables, which allows for conditional infer-
  ence by grounding macro-intents to manipulate behaviors.
- Focusing on multi-agent tracking data, we show that our approach can generate high-
  quality trajectories and effectively encode long-term coordination between multiple agents.

In addition to synthetic settings, we showcase our approach in an application on modeling team
offense in basketball. We validate our approach both quantitatively and qualitatively, including a
user study comparison with professional sports analysts, and show significant improvements over
standard baselines.

## 2 RELATED WORK

**Deep generative models.** The study of deep generative models is an increasingly popular research
area, due to their ability to inherit both the flexibility of deep learning and the probabilistic semantics
of generative models. In general, there are two ways that one can incorporate stochastics into deep
models. The first approach models an explicit distribution over actions in the output layer, e.g.,
via logistic regression (Chen et al., 2015; Oord et al., 2016a;b; Zheng et al., 2016; Eyjolfsdottir
et al., 2017). The second approach uses deep neural nets to define a transformation from a simple
distribution to one of interest (Goodfellow et al., 2014; Kingma & Welling, 2014; Rezende et al.,
2014) and can more readily be extended to incorporate additional structure, such as a hierarchy of
random variables (Ranganath et al., 2016) or dynamics (Johnson et al., 2016; Chung et al., 2015;
Krishnan et al., 2017; Fraccaro et al., 2016). Our framework can incorporate both variants.

**Structured probabilistic models.** Recently, there has been increasing interest in probabilistic mod-
eling with additional structure or side information. Existing work includes approaches that enforce
logic constraints (Akkaya et al., 2016), specify generative models as programs (Tran et al., 2016), or
automatically produce weak supervision via data programming (Ratner et al., 2016). Our framework
is inspired by the latter, which we extend to the spatiotemporal regime.

**Imitation Learning.** Our work is also related to imitation learning, which aims to learn a policy
that can mimic demonstrated behavior (Syed & Schapire, 2008; Abbeel & Ng, 2004; Ziebart et al.,
2008; Ho & Ermon, 2016). There has been some prior work in multi-agent imitation learning (Le
et al., 2017; Song et al., 2018) and learning stochastic policies (Ho & Ermon, 2016; Li et al., 2017),

but no previous work has focused on learning generative polices while simultaneously addressing generative and multi-agent imitation learning. For instance, experiments in (Ho & Ermon, 2016) all lead to highly peaked distributions, while (Li et al., 2017) captures multimodal distributions by learning unimodal policies for a fixed number of experts. (Hrolenok et al., 2017) raise the issue of learning stochastic multi-agent behavior, but their solution involves significant feature engineering.

## 3 BACKGROUND: SEQUENTIAL GENERATIVE MODELING

Let $\mathbf{x}_t \in \mathbb{R}^d$ denote the state at time $t$ and $\mathbf{x}_{\leq T} = \{\mathbf{x}_1, \ldots, \mathbf{x}_T\}$ denote a sequence of states of length $T$. Suppose we have a collection of $N$ demonstrations $\mathcal{D} = \{\mathbf{x}_{\leq T}\}$. In our experiments, all sequences have the same length $T$, but in general this does not need to be the case.

The goal of sequential generative modeling is to learn the distribution over sequential data $\mathcal{D}$. A common approach is to factorize the joint distribution and then maximize the log-likelihood:

$$\theta^* = \operatorname{argmax}_\theta \sum_{\mathbf{x}_{\leq T} \in \mathcal{D}} \log p_\theta(\mathbf{x}_{\leq T}) = \operatorname{argmax}_\theta \sum_{\mathbf{x}_{\leq T} \in \mathcal{D}} \sum_{t=1}^{T} \log p_\theta(\mathbf{x}_t | \mathbf{x}_{<t}), \quad (1)$$

where $\theta$ are the learn-able parameters of the model, such as a recurrent neural network (RNN).

**Stochastic latent variable models.** However, RNNs with simple output distributions that optimize Eq. (1) often struggle to capture highly variable and structured sequential data. For example, an RNN with Gaussian output distribution has difficulty learning the multimodal behavior of the green player moving to the top-left/bottom-left in Figure 1a. Recent work in sequential generative models address this issue by injecting stochastic latent variables into the model and optimizing using amortized variational inference to learn the latent variables (Fraccaro et al., 2016; Goyal et al., 2017).

In particular, we use a variational RNN (VRNN (Chung et al., 2015)) as our base model (Figure 3a), but we emphasize that our approach is compatible with other sequential generative models as well. A VRNN is essentially a variational autoencoder (VAE) conditioned on the hidden state of an RNN and is trained by maximizing the (sequential) evidence lower-bound (ELBO):

$$\mathbb{E}_{q_\phi(\mathbf{z}_{\leq T} | \mathbf{x}_{\leq T})} \left[ \sum_{t=1}^{T} \log p_\theta(\mathbf{x}_t \mid \mathbf{z}_{\leq t}, \mathbf{x}_{<t}) - D_{KL}\Big(q_\phi(\mathbf{z}_t \mid \mathbf{x}_{\leq t}, \mathbf{z}_{<t}) || p_\theta(\mathbf{z}_t \mid \mathbf{x}_{<t}, \mathbf{z}_{<t})\Big) \right]. \quad (2)$$

Eq. (2) is a lower-bound of the log-likelihood in Eq. (1) and can be interpreted as the VAE ELBO summed over each timestep $t$. We refer to appendix A for more details of VAEs and VRNNs.

## 4 HIERARCHICAL FRAMEWORK USING MACRO-INTENTS

In our problem setting, we assume that each sequence $\mathbf{x}_{\leq T}$ consists of the trajectories of $K$ coordinating agents. That is, we can decompose each $\mathbf{x}_{\leq T}$ into $K$ trajectories: $\mathbf{x}_{\leq T} = \{\mathbf{x}_{\leq T}^1, \ldots, \mathbf{x}_{\leq T}^K\}$. For example, the sequence in Figure 1a can be decomposed into the trajectories of $K = 5$ basketball players. Assuming conditional independence between the agent states $\mathbf{x}_t^k$ given state history $\mathbf{x}_{<t}$, we can factorize the maximum log-likelihood objective in Eq. (1) even further:

$$\theta^* = \operatorname{argmax}_\theta \sum_{\mathbf{x}_{\leq T} \in \mathcal{D}} \sum_{t=1}^{T} \sum_{k=1}^{K} \log p_{\theta_k}(\mathbf{x}_t^k | \mathbf{x}_{<t}). \quad (3)$$

Naturally, there are two baseline approaches in this setting:

1. Treat the data as a single-agent trajectory and train a single model: $\theta = \theta_1 = \cdots = \theta_K$.
2. Train independent models for each agent: $\theta = \{\theta_1, \ldots, \theta_K\}$.

As we empirically verify in Section 5, VRNN models using these two approaches have difficulty learning representations of the data that generalize well over long time horizons, and capturing the coordination inherent in multi-agent trajectories. Our solution introduces a hierarchical structure of *macro-intents* obtained via *labeling functions* to effectively learn low-dimensional (distributional) representations of the data that extend in both time and space for multiple coordinating agents.

**Defining macro-intents.** We assume there exists shared latent variables called macro-intents that: 1) provide a tractable way to capture coordination between agents; 2) encode long-term intents of agents and enable long-term planning at a higher-level timescale; and 3) compactly represent some low-dimensional structure in an exponentially large multi-agent state space.

For example, Figure 2 illustrates macro-intents for two basketball players as specific areas on the court (boxes). Upon reaching its macro-intent in the top-right, the blue player moves towards its next macro-intent in the bottom-left. Similarly, the green player moves towards its macro-intents from bottom-right to middle-left. These macro-intents are visible to both players and capture the coordination as they describe how the players plan to position themselves on the court. Macro-intents provide a compact summary of the players' trajectories over a long time.

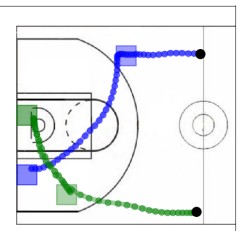

Figure 2: Macro-intents (boxes) for two players.

Macro-intents do not need to have a geometric interpretation. For example, macro-intents in the Boids model in Figure 1b can be a binary label indicating friendly vs. unfriendly behavior. The goal is for macro-intents to encode long-term intent and ensure that agents behave more cohesively. Our modeling assumptions for macro-intents are:

- agent states $\{\mathbf{x}_t^k\}$ in an episode $[t_1, t_2]$ are conditioned on some shared macro-intent $\mathbf{g}_t$,
- the start and end times $[t_1, t_2]$ of episodes can vary between trajectories,
- macro-intents change slowly over time relative to the agent states: $d\mathbf{g}_t/dt \ll 1$,
- and due to their reduced dimensionality, we can model (near-)arbitrary dependencies between macro-intents (e.g., coordination) via black box learning.

**Labeling functions for macro-intents.** Obtaining macro-intent labels from experts for training is ideal, but often too expensive. Instead, our work is inspired by recent advances in weak supervision settings known as *data programming*, in which multiple weak and noisy label sources called labeling functions can be leveraged to learn the underlying structure of large unlabeled datasets (Ratner et al., 2018; Bach et al., 2017). These labeling functions often compute heuristics that allow users to incorporate domain knowledge into the model. For instance, the labeling function we use to obtain macro-intents for basketball trajectories computes the regions on the court in which players remain stationary; this integrates the idea that players aim to set up specific formations on the court. In general, labeling functions are simple scripts/programs that can parse and label data very quickly, hence the name *programmatic weak supervision*.

Other approaches that try to learn macro-intents in a fully unsupervised learning setting can encounter difficulties that have been previously noted, such as the importance of choosing the correct prior and approximate posterior (Rezende & Mohamed, 2015) and the interpretability of learned latent variables (Chen et al., 2017). We find our approach using labeling functions to be much more attractive, as it outperforms other baselines by generating samples of higher quality, while also avoiding the engineering required to address the aforementioned difficulties.

**Hierarchical model with macro-intents** Our hierarchical model uses an intermediate layer to model macro-intent, so our agent VRNN-models becomes:

$$p_{\theta_k}(\mathbf{x}_t^k | \mathbf{x}_{<t}) = \varphi^k(\mathbf{z}_t^k, \mathbf{h}_{t-1}^k, \mathbf{g}_t), \tag{4}$$

where $\varphi^k$ maps to a distribution over states, $\mathbf{z}_t^k$ is the VRNN latent variable, $\mathbf{h}_t^k$ is the hidden state of an RNN that summarizes the trajectory up to time $t$, and $\mathbf{g}_t$ is the shared macro-intent at time $t$. Figure 3b shows our hierarchical model, which samples macro-intents during generation rather than using only ground-truth macro-intents. Here, we train an RNN-model to sample macro-intents:

$$p(\mathbf{g}_t | \mathbf{g}_{<t}) = \varphi_g(\mathbf{h}_{g,t-1}, \mathbf{x}_{t-1}), \tag{5}$$

where $\varphi^g$ maps to a distribution over macro-intents and $\mathbf{h}_{g,t-1}$ summarizes the history of macro-intents up to time $t$. We condition the macro-intent model on previous states $\mathbf{x}_{t-1}$ in Eq. (5) and generate next states by first sampling a macro-intent $\mathbf{g}_t$, and then sampling $\mathbf{x}_t^k$ conditioned on $\mathbf{g}_t$ (see Figure 3b). Note that all agent-models for generating $\mathbf{x}_t^k$ share the same macro-intent variable $\mathbf{g}_t$. This is core to our approach as it induces coordination between agent trajectories (see Section 5).

We learn our agent-models by maximizing the VRNN objective from Eq (2) conditioned on the shared $\mathbf{g}_t$ variables while independently learning the macro-intent model via supervised learning by maximizing the log-likelihood of macro-intent labels obtained programmatically.

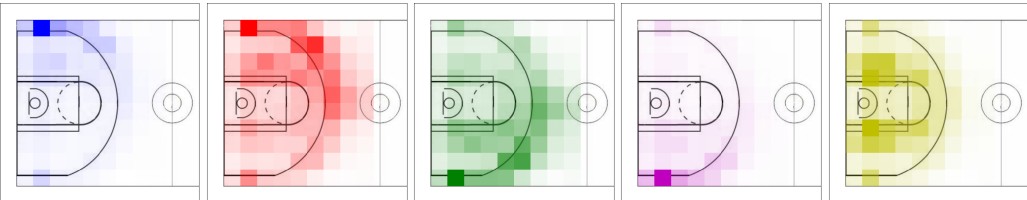

Figure 4: Distribution of weak macro-intent labels extracted for each player from the training data. Color intensity corresponds to frequency of macro-intent label. Players are ordered by their relative positions on the court, which can be seen from the macro-intent distributions.

## 5 EXPERIMENTS

We first apply our approach on generating offensive team basketball gameplay (team with possession of the ball), and then on a synthetic Boids model dataset. We present both quantitative and qualitative experimental results. Our quantitative results include a user study comparison with professional sports analysts, who significantly preferred basketball rollouts generated from our approach to standard baselines. Examples from the user study and videos of generated rollouts can be seen in our demo video.[2] Our qualitative results demonstrate the ability of our approach to generate high-quality rollouts under various conditions.

### 5.1 EXPERIMENTAL SETUP FOR BASKETBALL

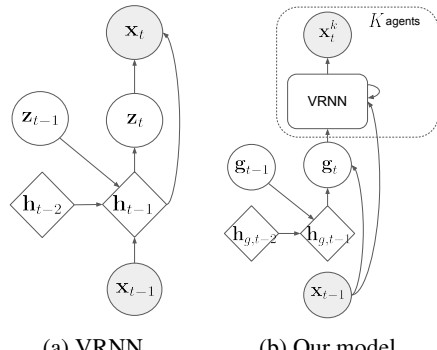

(a) VRNN      (b) Our model

Figure 3: Depicting VRNN and our model. Circles are stochastic and diamonds are deterministic. macro-intent $\mathbf{g}_t$ is shared across agents. In principle, any generative model can be used in our framework.

**Training data.** Each demonstration in our data contains trajectories of $K = 5$ players on the left half-court, recorded for $T = 50$ timesteps at 6 Hz. The offensive team has possession of the ball for the entire sequence. $\mathbf{x}_t^k$ are the coordinates of player $k$ at time $t$ on the court ($50 \times 94$ feet). We normalize and mean-shift the data. Players are ordered based on their relative positions, similar to the role assignment in (Lucey et al., 2013). There are 107,146 training and 13,845 test examples. We ignore the defensive players and the ball to focus on capturing the coordination and multimodality of the offensive team. In principle, we can provide the defensive positions as conditional input for our model and update the defensive positions using methods such as (Le et al., 2017). We leave the task of modeling the ball and defense for future work.

**Macro-intent labeling function.** We extract weak macro-intent labels $\hat{\mathbf{g}}_t^k$ for each player $k$ as done in (Zheng et al., 2016). We segment the left half-court into a $10 \times 9$ grid of 5ft ×5ft boxes. The weak macro-intent $\hat{\mathbf{g}}_t^k$ at time $t$ is a 1-hot encoding of dimension 90 of the next box in which player $k$ is stationary (speed $\|\mathbf{x}_{t+1}^k - \mathbf{x}_t^k\|_2$ below a threshold). The shared global macro-intent $\mathbf{g}_t$ is the concatenation of individual macro-intents. Figure 4 shows the distribution of macro-intents for each player. We refer to this labeling function as `LF-stationary` (pseudocode in appendix D).

**Model details.** We model each latent variable $\mathbf{z}_t^k$ as a multivariate Gaussian with diagonal covariance of dimension 16. All output models are implemented with memory-less 2-layer fully-connected neural networks with a hidden layer of size 200. Our agent-models sample from a multivariate Gaussian with diagonal covariance while our macro-intent models sample from a multinomial distribution over the macro-intents. All hidden states ($\mathbf{h}_{g,t}, \mathbf{h}_t^1, \dots \mathbf{h}_t^K$) are modeled with 200 2-layer GRU memory cells each. We maximize the log-likelihood/ELBO with stochastic gradient descent using the Adam optimizer (Kingma & Ba, 2015) and a learning rate of 0.0001.

**Baselines.** We compare with 5 baselines that do not use macro-intents from labeling functions:

---

[2]Demo video: https://youtu.be/0q1j22yMipY

| MODEL | BASKETBALL | BOIDS |
|---|---|---|
| RNN-GAUSS | 1931 | 2414 |
| VRNN-SINGLE | $\geq 2302$ | $\geq 2417$ |
| VRNN-INDEP | $\geq 2360$ | $\geq 2385$ |
| VRNN-MIXED | $\geq 2323$ | $\geq 2204$ |
| VRAE-MI | $\geq 2349$ | $\geq 2331$ |
| OURS | $\geq \mathbf{2362}$ | $\geq \mathbf{2428}$ |

Table 1: Average log-likelihoods per test sequence. "$\geq$" indicates ELBO of log-likelihood. Our hierarchical model achieves higher log-likelihoods than baselines for both datasets.

| VS. MODEL | WIN/TIE/LOSS | AVG GAIN |
|---|---|---|
| VS. VRNN-SINGLE | 25/0/0 | 0.57 |
| VS. VRNN-INDEP | 15/4/6 | 0.23 |

Table 2: Basketball preference study results. Win/Tie/Loss indicates how often our model is preferred over baselines (25 comparisons per baseline). Gain is computed by scoring +1 when our model is preferred and -1 otherwise. Results are 98% significant using a one-sample t-test.

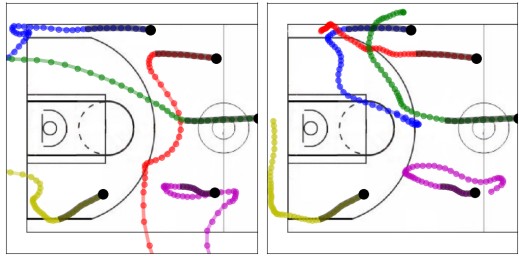

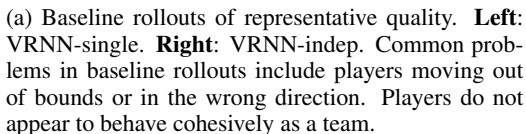

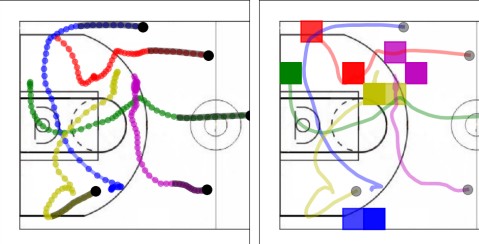

(a) Baseline rollouts of representative quality. **Left**: VRNN-single. **Right**: VRNN-indep. Common problems in baseline rollouts include players moving out of bounds or in the wrong direction. Players do not appear to behave cohesively as a team.

(b) **Left**: Rollout from our model. All players remain in bounds. **Right**: Corresponding macro-intents for left rollout. Macro-intent generation is stable and suggests that the team is creating more space for the blue player (perhaps setting up an isolation play).

Figure 5: Rollouts from baselines and our model starting from black dots, generated for 40 timesteps after an initial burn-in period of 10 timesteps (marked by dark shading). An interactive demo of our hierarchical model is available at: `http://basketball-ai.com/`.

1. **RNN-gauss:** RNN without latent variables using 900 2-layer GRU cells as hidden state.

2. **VRNN-single:** VRNN in which we concatenate all player positions together ($K = 1$) with 900 2-layer GRU cells for the hidden state and a 80-dimensional latent variable.

3. **VRNN-indep:** VRNN for each agent with 250 2-layer GRUs and 16-dim latent variables.

4. **VRNN-mixed:** Combination of VRNN-single and VRNN-indep. Shared hidden state of 600 2-layer GRUs is fed into decoders with 16-dim latent variables for each agent.

5. **VRAE-mi:** VRAE-style architecture (Fabius & van Amersfoort, 2014) that maximizes the mutual information between $\mathbf{x}_{\leq T}$ and macro-intent. We refer to appendix C for details.

## 5.2 QUANTITATIVE EVALUATION FOR BASKETBALL

**Log-likelihood.** Table 1 reports the average log-likelihoods on the test data. Our approach outperforms RNN-gauss and is comparable with other baselines. However, higher log-likelihoods do not necessarily indicate higher quality of generated samples (Theis et al., 2015). As such, we also assess using other means, such as human preference studies and auxiliary statistics.

**Human preference study.** We recruited 14 professional sports analysts as judges to compare the quality of rollouts. Each comparison animates two rollouts, one from our model and another from a baseline. Both rollouts are burned-in for 10 timesteps with the same ground-truth states from the test set, and then generated for the next 40 timesteps. Judges decide which of the two rollouts looks more realistic. Table 2 shows the results from the preference study. We tested our model against two baselines, VRNN-single and VRNN-indep, with 25 comparisons for each. All judges preferred our model over the baselines with 98% statistical significance. These results suggest that our model generates rollouts of significantly higher quality than the baselines.

**Domain statistics.** Finally, we compute several basketball statistics (average speed, average total distance traveled, % of frames with players out-of-bounds) and summarize them in Table 3. Our

| Model | Speed (ft) | Distance (ft) | OOB (%) |
|---|---|---|---|
| RNN-gauss | 3.05 | 149.57 | 46.93 |
| VRNN-single | 1.28 | 62.67 | 45.67 |
| VRNN-indep | 0.89 | 43.78 | 33.78 |
| VRNN-mixed | 0.91 | 44.80 | 27.19 |
| VRAE-mi | 0.98 | 48.25 | 20.09 |
| Ours (LF-window50) | 0.99 | 48.53 | 28.84 |
| Ours (LF-window25) | 0.87 | 42.99 | **14.53** |
| **Ours (LF-stationary)** | **0.79** | **38.92** | 15.52 |
| Ground-truth | 0.77 | 37.78 | 2.21 |

Table 3: Domain statistics of 1000 basketball trajectories generated from each model: average speed, average distance traveled, and % of frames with players out-of-bounds (OOB). Trajectories from our models using programmatic weak supervision match the closest with the ground-truth. See appendix D for labeling function pseudocode.

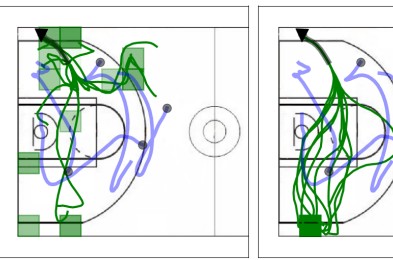 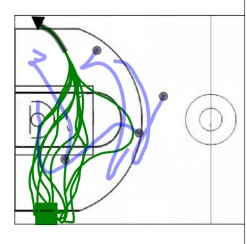 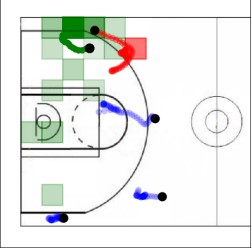 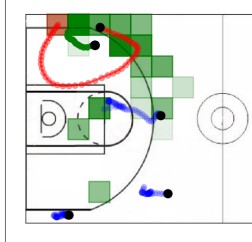

(a) 10 rollouts of the green player (▼) with a burn-in period of 20 timesteps. **Left**: The model generates macro-intents. **Right**: We ground the macro-intents at the bottom-left. In both, we observe a multimodal distribution of trajectories.

(b) The distribution of macro-intents sampled from 20 rollouts of the green player changes in response to the change in red trajectories and macro-intents. This suggests that macro-intents encode and induce coordination between multiple players.

Figure 6: Rollouts from our model demonstrating the effectiveness of macro-intents in generating coordinated multi-agent trajectories. Blue trajectories are fixed and (●) indicates initial positions.

model generates trajectories that are most similar to ground-truth trajectories with respect to these statistics, indicating that our model generates significantly more realistic behavior than all baselines.

**Choice of labeling function.** In addition to `LF-stationary`, we also assess the quality of our approach using macro-intents obtained from different labeling functions. `LF-window25` and `LF-window50` labels macro-intents as the last region a player resides in every window of 25 and 50 timesteps respectively (pseudocode in appendix D). Table 3 shows that domain statistics from our models using programmatic weak supervision match closer to the ground-truth with more informative labeling functions (`LF-stationary` > `LF-window25` > `LF-window50`). This is expected, since `LF-stationary` provides the most information about the structure of the data.

### 5.3 QUALITATIVE EVALUATION OF GENERATED ROLLOUTS FOR BASKETBALL

We next conduct a qualitative visual inspection of rollouts. Figure 5 shows rollouts generated from VRNN-single, VRNN-indep, and our model by sampling states for 40 timesteps after an initial burn-in period of 10 timesteps with ground-truth states from the test set. An interactive demo to generate more rollouts from our hierarchical model can be found at: `http://basketball-ai.com/`.

Common problems in baseline rollouts include players moving out of bounds or in the wrong direction (Figure 5a). These issues tend to occur at later timesteps, suggesting that the baselines do not perform well over long horizons. One possible explanation is due to compounding errors (Ross et al., 2011): if the model makes a mistake and deviates from the states seen during training, it is likely to make more mistakes in the future and generalize poorly. On the other hand, generated rollouts from our model are more robust to the types of errors made by the baselines (Figure 5b).

**Macro-intents induce multimodal and interpretable rollouts.** Generated macro-intents allow us to intepret the intent of each individual player as well as a global team strategy (e.g. setting up a specific formation on the court). We highlight that our model learns a multimodal generating distribution, as repeated rollouts with the same burn-in result in a dynamic range of generated trajectories, as seen in Figure 6a Left. Furthermore, Figure 6a Right demonstrates that grounding macro-intents during generation instead of sampling them allows us to control agent behavior.

Figure 7: Synthetic Boids experiments. Showing histograms (horizontal axis: distance; vertical: counts) of average distance to an agent's closest neighbor in 5000 rollouts. Our hierarchical model more closely captures the two distinct modes for friendly (small distances, left peak) vs. unfriendly (large distances, right peak) behavior compared to baselines, which do not learn to distinguish them.

**Macro-intents induce coordination.** Figure 6b illustrates how the macro-intents encode coordination between players that results in realistic rollouts of players moving cohesively. As we change the trajectory and macro-intent of the red player, the distribution of macro-intents generated from our model for the green player changes such that the two players occupy different areas of the court.

## 5.4 SYNTHETIC EXPERIMENTS: BOIDS MODEL OF SCHOOLING BEHAVIOR

To illustrate the generality of our approach, we apply our model to a simplified version of the Boids model (Reynolds, 1987) that produces realistic trajectories of schooling behavior. We generate trajectories for 8 agents for 50 frames. The agents start in fixed positions around the origin with initial velocities sampled from a unit Gaussian. Each agent's velocity is then updated at each timestep:

$$\mathbf{v}_{t+1} = \beta \mathbf{v}_t + \beta(c_1 \mathbf{v}_{\text{coh}} + c_2 \mathbf{v}_{\text{sep}} + c_3 \mathbf{v}_{\text{ali}} + c_4 \mathbf{v}_{\text{ori}}). \tag{6}$$

Full details of the model can be found in Appendix B. We randomly sample the sign of $c_1$ for each trajectory, which produces two distinct types of behaviors: *friendly agents* ($c_1 > 0$) that like to group together, and *unfriendly agents* ($c_1 < 0$) that like to stay apart (see Figure 1b). We also introduce more stochasticity into the model by periodically updating $\beta$ randomly.

Our labeling function thresholds the average distance to an agent's closest neighbor (see last plot in Figure 7). This is equivalent to using the sign of $c_1$ as our macro-intents, which indicates the type of behavior. Note that unlike our macro-intents for the basketball dataset, these macro-intents are simpler and have no geometric interpretation. All models have similar average log-likelihoods on the test set in Table 1, but our hierarchical model can capture the true generating distribution much better than the baselines. For example, Figure 7 depicts the histograms of average distances to an agent's closest neighbor in trajectories generated from all models and the ground-truth. Our model more closely captures the two distinct modes in the ground-truth (friendly, small distances, left peak vs. unfriendly, large distances, right peak) whereas the baselines fail to distinguish them.

## 5.5 INSPECTING THE HIERARCHICAL MODEL CLASS

**Output distribution for states.** The outputs of all models (including baselines) sample from a multivariate Gaussian with diagonal covariance. We also experimented with sampling from a mixture of 2, 3, 4, and 8 Gaussian components, but discovered that the models would always learn to assign all the weight on a single component and ignore the others. The variance of the active component is also very small. This is intuitive because sampling with a large variance at every timestep would result in noisy trajectories and not the smooth ones that we see in Figures 5, 6a.

**Choice of macro-intent model.** In principle, we can use more expressive generative models, like a VRNN, to model macro-intents over richer macro-intent spaces in Eq. (5). In our case, we found that an RNN was sufficient in capturing the distribution of macro-intents shown in Figure 4. The RNN learns multinomial distributions over macro-intents that are peaked at a single macro-intent and relatively static through time, which is consistent with the macro-intent labels that we extracted from data. Latent variables in a VRNN had minimal effect on the multinomial distribution.

**Maximizing mutual information isn't effective.** The learned macro-intents in our fully unsupervised VRAE-mi model do not encode anything useful and are essentially ignored by the model. In particular, the model learns to match the approximate posterior of macro-intents from the encoder with the discriminator from the mutual information lower-bound. This results in a lack of diversity in rollouts as we vary the macro-intents during generation. We refer to appendix C for examples.

## 6 DISCUSSION

The macro-intents labeling functions used in our experiments are relatively simple. For instance, rather than simply using location-based macro-intents, we can also incorporate complex interactions such as "pick and roll". Another future direction is to explore how to adapt our method to different domains, e.g., defining a macro-intent representing "argument" for a dialogue between two agents, or a macro-intent representing "refrain" for music generation for "coordinating instruments" (Thickstun et al., 2017). We have shown that weak macro-intent labels extracted using simple domain-specific heuristics can be effectively used to generate high-quality coordinated multi-agent trajectories. An interesting direction is to incorporate multiple labeling functions, each viewed as noisy realizations of true macro-intents, similar to (Ratner et al., 2016; 2018; Bach et al., 2017).

ACKNOWLEDGMENTS

This research is supported in part by NSF #1564330, NSF #1637598, and gifts from Bloomberg, Activision/Blizzard and Northrop Grumman. Dataset was provided by STATS: `https://www.stats.com/data-science/`.

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

## A  SEQUENTIAL GENERATIVE MODELS

**Recurrent neural networks.**  A RNN models the conditional probabilities in Eq. (1) with a hidden state $\mathbf{h}_t$ that summarizes the information in the first $t - 1$ timesteps:

$$p_\theta(\mathbf{x}_t|\mathbf{x}_{<t}) = \varphi(\mathbf{h}_{t-1}), \qquad \mathbf{h}_t = f(\mathbf{x}_t, \mathbf{h}_{t-1}), \tag{7}$$

where $\varphi$ maps the hidden state to a probability distribution over states and $f$ is a deterministic function such as LSTMs (Hochreiter & Schmidhuber, 1997) or GRUs (Cho et al., 2014). RNNs with simple output distributions often struggle to capture highly variable and structured sequential data. Recent work in sequential generative models address this issue by injecting stochastic latent variables into the model and using amortized variational inference to infer latent variables from data.

**Variational Autoencoders.**  A variational autoencoder (VAE) (Kingma & Welling, 2014) is a generative model for non-sequential data that injects latent variables $\mathbf{z}$ into the joint distribution $p_\theta(\mathbf{x}, \mathbf{z})$ and introduces an inference network parametrized by $\phi$ to approximate the posterior $q_\phi(\mathbf{z} \mid \mathbf{x})$. The learning objective is to maximize the evidence lower-bound (ELBO) of the log-likelihood with respect to the model parameters $\theta$ and $\phi$:

$$\mathbb{E}_{q_\phi(\mathbf{z}|\mathbf{x})} \left[ \log p_\theta(\mathbf{x}|\mathbf{z}) \right] - D_{KL}(q_\phi(\mathbf{z} \mid \mathbf{x})||p_\theta(\mathbf{z})) \tag{8}$$

The first term is known as the reconstruction term and can be approximated with Monte Carlo sampling. The second term is the Kullback-Leibler divergence between the approximate posterior and the prior, and can be evaluated analytically (i.e. if both distributions are Gaussian with diagonal covariance). The inference model $q_\phi(\mathbf{z} \mid \mathbf{x})$, generative model $p_\theta(\mathbf{x} \mid \mathbf{z})$, and prior $p_\theta(\mathbf{z})$ are often implemented with neural networks.

**Variational RNNs.**  VRNNs combine VAEs and RNNs by conditioning the VAE on a hidden state $\mathbf{h}_t$ (see Figure 3a):

$$\begin{align}
p_\theta(\mathbf{z}_t|\mathbf{x}_{<t}, \mathbf{z}_{<t}) &= \varphi_{\text{prior}}(\mathbf{h}_{t-1}) & \text{(prior)} \tag{9}\\
q_\phi(\mathbf{z}_t|\mathbf{x}_{\leq t}, \mathbf{z}_{<t}) &= \varphi_{\text{enc}}(\mathbf{x}_t, \mathbf{h}_{t-1}) & \text{(inference)} \tag{10}\\
p_\theta(\mathbf{x}_t|\mathbf{z}_{\leq t}, \mathbf{x}_{<t}) &= \varphi_{\text{dec}}(\mathbf{z}_t, \mathbf{h}_{t-1}) & \text{(generation)} \tag{11}\\
\mathbf{h}_t &= f(\mathbf{x}_t, \mathbf{z}_t, \mathbf{h}_{t-1}). & \text{(recurrence)} \tag{12}
\end{align}$$

VRNNs are also trained by maximizing the ELBO, which in this case can be interpreted as the sum of VAE ELBOs over each timestep of the sequence:

$$\mathbb{E}_{q_\phi(\mathbf{z}_{\leq T}|\mathbf{x}_{\leq T})} \left[ \sum_{t=1}^{T} \log p_\theta(\mathbf{x}_t \mid \mathbf{z}_{\leq T}, \mathbf{x}_{<t}) - D_{KL}\Big( q_\phi(\mathbf{z}_t \mid \mathbf{x}_{\leq T}, \mathbf{z}_{<t})||p_\theta(\mathbf{z}_t \mid \mathbf{x}_{<t}, \mathbf{z}_{<t}) \Big) \right] \tag{13}$$

Note that the prior distribution of latent variable $\mathbf{z}_t$ depends on the history of states and latent variables (Eq. (9)). This temporal dependency of the prior allows VRNNs to model complex sequential data like speech and handwriting (Chung et al., 2015).

## B  BOIDS MODEL DETAILS

We generate 32,768 training and 8,192 test trajectories. Each agent's velocity is updated as:

$$\mathbf{v}_{t+1} = \beta \mathbf{v}_t + \beta(c_1 \mathbf{v}_{\text{coh}} + c_2 \mathbf{v}_{\text{sep}} + c_3 \mathbf{v}_{\text{ali}} + c_4 \mathbf{v}_{\text{ori}}), \tag{14}$$

- $\mathbf{v}_{\text{coh}}$ is the normalized cohesion vector towards an agent's local neighborhood (radius 0.9)
- $\mathbf{v}_{\text{sep}}$ is the normalized vector away from an agent's close neighborhood (radius 0.2)
- $\mathbf{v}_{\text{ali}}$ is the average velocity of other agents in a local neighborhood
- $\mathbf{v}_{\text{ori}}$ is the normalized vector towards the origin
- $(c_1, c_2, c_3, c_4) = (\pm 1, 0.1, 0.2, 1)$
- $\beta$ is sampled uniformly at random every 10 frames in range $[0.8, 1.4]$

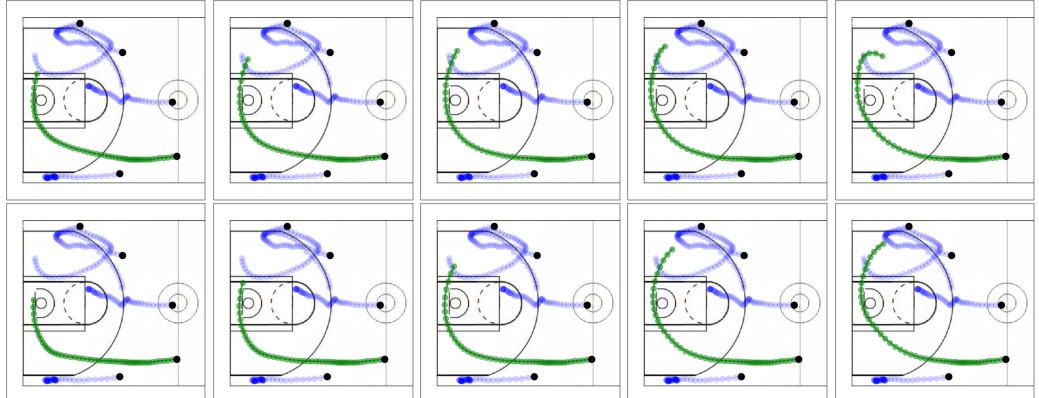

Figure 8: Average distribution of 8-dimensional categorical macro-intent variable. The encoder and discriminator distributions match, but completely ignore the uniform prior distribution.

Figure 9: Generated trajectories of green player conditioned on fixed blue players given various 2-dimensional macro-intent variables with a standard Gaussian prior. **Left to Right columns**: values of 1st dimension in $\{-1, -0.5, 0, 0.5, 1\}$. **Top row**: 2nd dimension equal to $-0.5$. **Bottom row**: 2nd dimension equal to $0.5$. We see limited variability as we change the macro-intent variable.

## C   MAXIMIZING MUTUAL INFORMATION

We ran experiments to see if we can learn meaningful macro-intents in a fully unsupervised fashion by maximizing the mutual information between macro-intent variables and trajectories $\mathbf{x}_{\leq T}$. We use a VRAE-style model from (Fabius & van Amersfoort, 2014) in which we encode an entire trajectory into a latent macro-intent variable $\mathbf{z}$, with the idea that $\mathbf{z}$ should encode global properties of the sequence. The corresponding ELBO is:

$$\mathcal{L}_1 = \mathbb{E}_{q_\phi(\mathbf{z}|\mathbf{x}_{\leq T})}\left[\sum_{t=1}^{T}\sum_{k=1}^{K}\log p_{\theta_k}(\mathbf{x}_t^k \mid \mathbf{x}_{<t}, \mathbf{z})\right] - D_{KL}\Big(q_\phi(\mathbf{z} \mid \mathbf{x}_{\leq T})||p_\theta(\mathbf{z})\Big), \qquad (15)$$

where $p_\theta(\mathbf{z})$ is the prior, $q_\phi(\mathbf{z} \mid \mathbf{x}_{\leq T})$ is the encoder, and $p_{\theta_k}(\mathbf{x}_t^k \mid \mathbf{x}_{<t}, \mathbf{z})$ are decoders per agent.

It is intractable to compute the mutual information between $\mathbf{z}$ and $\mathbf{x}_{\leq T}$ exactly, so we introduce a discriminator $q_\psi(\mathbf{z} \mid \mathbf{x}_{\leq T})$ and use the following variational lower-bound of mutual information:

$$\mathcal{L}_2 = \mathcal{H}(\mathbf{z}) + \mathbb{E}_{p_\theta(\mathbf{x}_{\leq T}|\mathbf{z})}\Big[\mathbb{E}_{q_\phi(\mathbf{z}|\mathbf{x}_{\leq T})}\big[\log q_\psi(\mathbf{z} \mid \mathbf{x}_{\leq T})\big]\Big] \leq MI(\mathbf{x}_{\leq T}, \mathbf{z}). \qquad (16)$$

We jointly maximize $\mathcal{L}_1 + \lambda\mathcal{L}_2$ wrt. model parameters $(\theta, \phi, \psi)$, with $\lambda = 1$ in our experiments.

**Categorical vs. real-valued macro-intent z.**   When we train an 8-dimensional categorical macro-intent variable with a uniform prior (using gumbel-softmax trick (Jang et al., 2017)), the average distribution from the encoder matches the discriminator but not the prior (Figure 8). When we train a 2-dimensional real-valued macro-intent variable with a standard Gaussian prior, the learned model generates trajectories with limited variability as we vary the macro-intent variable (Figure 9).

# D LABELING FUNCTIONS FOR MACRO-INTENTS IN BASKETBALL

We define macro-intents in basketball by segmenting the left half-court into a $10 \times 9$ grid of 5ft $\times$5ft boxes (Figure 2). Algorithm 1 describes `LF-window25`, which computes macro-intents based on last positions in 25-timestep windows (`LF-window50` is similar). Algorithm 2 describes `LF-stationary`, which computes macro-intents based on stationary positions. For both, `Label-macro-intent`$(\mathbf{x}_t^k)$ returns the 1-hot encoding of the box that contains the position $\mathbf{x}_t^k$.

---

**Algorithm 1** Labeling function that computes macro-intents in 25-timestep windows

---

1: **procedure** LF-WINDOW25($\mathbf{x}_{\leq T}$)         $\triangleright$ Trajectory $\mathbf{x}_{\leq T}$ of $K$ players
2:      macro-intents $\mathbf{g} \leftarrow$ initialize array of size $(K, T, 90)$
3:      **for** $k = 1 \ldots K$ **do**
4:         $\mathbf{g}[k, T] \leftarrow$ LABEL-MACRO-INTENT($\mathbf{x}_T^k$)         $\triangleright$ Last timestep
5:         **for** $t = T - 1 \ldots 1$ **do**
6:            **if** (t+1) mod 25 == 0 **then**         $\triangleright$ End of 25-timestep window
7:               $\mathbf{g}[k, t] \leftarrow$ LABEL-MACRO-INTENT($\mathbf{x}_t^k$)
8:            **else**
9:               $\mathbf{g}[k, t] \leftarrow \mathbf{g}[k, t + 1]$
10:      **return g**

---

**Algorithm 2** Labeling function that computes macro-intents based on stationary positions

---

1: **procedure** LF-STATIONARY($\mathbf{x}_{\leq T}$)         $\triangleright$ Trajectory $\mathbf{x}_{\leq T}$ of $K$ players
2:      macro-intents $\mathbf{g} \leftarrow$ initialize array of size $(K, T, 90)$
3:      **for** $k = 1 \ldots K$ **do**
4:         speed $\leftarrow$ compute speeds of player $k$ in $\mathbf{x}_{\leq T}^k$
5:         stationary $\leftarrow$ speed $<$ threshold
6:         $\mathbf{g}[k, T] \leftarrow$ LABEL-MACRO-INTENT($\mathbf{x}_T^k$)         $\triangleright$ Last timestep
7:         **for** $t = T - 1 \ldots 1$ **do**
8:            **if** stationary[t] and not stationary[t+1] **then**         $\triangleright$ Player $k$ starts moving
9:               $\mathbf{g}[k, t] \leftarrow$ LABEL-MACRO-INTENT($\mathbf{x}_t^k$)
10:           **else**         $\triangleright$ Player $k$ remains stationary
11:               $\mathbf{g}[k, t] \leftarrow \mathbf{g}[k, t + 1]$
12:      **return g**

---

