# OpenReview forum: "Generating Multi-Agent Trajectories using Programmatic Weak Supervision"
_ICLR.cc/2019/Conference_

### Official Review · AnonReviewer3 · 2018-11-04
**Heuristic labeling enables learning of hierarchical model without needing to marginalize over latent variables**

**Rating:** 6
**Confidence:** 3

**Review:**

# Summary

The paper proposes training generative models that produce multi-agent trajectories using heuristic functions that label variables that would otherwise be latent in training data. The generative models are hierarchical, and these latent variables correspond to higher level goals in agent behavior. The paper focuses on basketball offenses as a motivating scenario in which multiple agents have coordinated high-level behavior. The generative models are RNNs where each output is fed into the decoder of a variational autoencoder to produce observed states. The authors add an intermediate layer to capture the latent variables, called macro-intents. The parameters are learned by maximizing an evidence lower bound.

Experiments qualitatively and quantitatively show that the hierarchical model produces realistic multi-agent traces.

# Comments

The paper presents a sensible solution for heuristically labeling latent variables. It is not particularly surprising that the model then learns useful behavior because it no longer has to maximize the marginal likelihood over all possible macro-intents. What is more interesting is that a heuristic labeling function is sufficient to label macro-intents that lead to learning realistic basketball offenses and swarm behavior.

Are any of the baselines (VRNN-single, VRNN-indep, and VRNN-mi) equivalent to training the hierarchical model by maximizing an ELBO on the marginal likelihood? I do not think this comparison is done, which might be interesting to quantify how much of a difference heuristic labeling makes. Of course, the potentially poor fit of a variational distribution would confound the results.

# Minor things

1) In the caption of Table 1, it says "Our hierarchical model achieves higher log-likelihoods than baselines for both datasets." Are not the reported scores evidence lower-bounds? So it achieves a higher evidence lower bound, but without actually computing the true likelihood, could not the other models have higher likelihoods?

2) Under "Human preference study" it says "All judges preferred our model over the baselines with 98% statistical significance." I am not familiar with this terminology. Does that mean that a p value for some null hypothesis is .02?

3) Something is wrong with the citation commands. Perhaps \citep should be used.

---

> ### Author Response · Authors · 2018-11-20
> **AnonReviewer3 Response**
>
> Thank you for reviewing our paper and providing insightful feedback. We respond to your main points below.
>
> > “What is more interesting is that a heuristic labeling function is sufficient to label macro-intents that lead to learning realistic basketball offense and swarm behavior.”
>
> Yes, we are very excited at the new lines of research that this opens up. One can envision many settings in which users wish to have diverse and detailed control over what’s being generated. We believe models with this degree of control can be learned by incorporating labeling functions defined by users according to their preferences. We are very excited about future work in this direction.
>
> > “Are any … baselines … equivalent to training the hierarchical model by maximizing an ELBO on the marginal likelihood?”
>
> If we understand the reviewer’s question, then VRAE-mi (previously named VRNN-mi) does exactly this by introducing a global latent variable (in place of macro-intent weak labels) and maximizing the ELBO as well as the mutual information between the global latent variable and the trajectory. We will update the paper to make this more clear.
>
> > “... could not the other models have higher likelihoods?”
>
> Yes, a higher ELBO does not imply a higher true likelihood, as it depends on the tightness of the bound. Computing the exact likelihood is infeasible, but it can be approximated with importance sampling. However, we note that likelihoods do not necessary correspond to quality of generated samples, as evidenced by our experiments and by [1]. Furthermore, reporting ELBOs is often sufficient when quantitatively comparing models [2,3,4].
>
> “> ... 98% statistical significance.”
>
> We performed a one-sample t-test, where the null hypothesis is that the gains come from a zero-mean distribution (which would mean that both models are preferred equally).
>
> [1] Theis et al. A note on the evaluation of generative models.
> [2] Chung et al. A recurrent latent variable model for sequential data.
> [3] Fraccaro et al. Sequential neural models with stochastic layers.
> [4] Goyal et al. Z-forcing: training stochastic recurrent networks.

---

### Official Review · AnonReviewer2 · 2018-11-09
**Hierarchical latent variables with weak supervision help learning a global coordination between cooperative agents.**

**Rating:** 6
**Confidence:** 3

**Review:**


This paper proposes training multiple generative models that share a common latent variable, which is learned in a weakly supervised fashion, to achieve high level coordination between multiple agents. Each agent has a separate VRNN model which is conditioned on the agent’s own trajectory history as well as the shared latent variable. The model is trained to maximize the ELBO objective and log-likelihood over macro-intent labels. Experimental results are conducted over a basketball gameplay dataset (to model the trajectories of the offensive team members) and a synthetic dataset. The results show that the proposed model is on-par with the baseline models in terms of ELBO while showing that it can model multi-modality better and is preferred more by humans.

In general, the paper is well written and the overall framework captures the essence of the problem that the authors are trying to solve.
Furthermore, incorporating an auxiliary latent variable to model the coordination between multiple agents is interesting.
I have several comments related to the strength of the baselines and contribution of individual components in the proposed model.


Major Comments

- It seems that VRNN-single and VRNN-indep are two models on the far two ends of a spectrum. To understand the contribution of the shared macro-intent, how would an intermediate baseline model where a set of parameters are shared between agents and each agent also has an independent set of parameters perform? This could be accomplished by sharing the parameters of the first layer of GRU networks and learning the second layer parameters independently.

- How is the threshold for macro-intent generation selected? How does this parameter affect the overall performance? Since the smoothness of the segments between two macro-intents depend on this parameter, I am wondering its effect on the learned posterior distribution.

- Rather than using the prediction of the macro-intent RNN as a single global vector (\hat{g}_t), could using separate vectors for each agent (corresponding blocks of \hat{g}_t) as inputs to VRNN give the same results? Since the macro-intent RNN is already aware of all the macro-intents, it would be interesting to see if individual macro-intents are sufficient for VRNN to generate corresponding trajectories.


Minor Comments

- Do results in Table (1) come from sampling or using mode of the distributions? How peaked are the learned posterior distributions?
- What is the performance of the macro-intent RNN model?
- In Eq (2), “<=T” should be “<=t” (as in Eq (11) in Chung 2015).
- In Page 6, bullet point 4: it should be “except we maximize the mutual information…”

---

> ### Author Response · Authors · 2018-11-20
> **AnonReviewer2 Response**
>
> Thank you for reviewing our paper and providing insightful feedback. We respond to your main points below.
>
> > “... how would an intermediate baseline model where a set of parameters are shared and each agent also has an independent set of parameters perform?”
>
> Following your suggestion, we trained such a model where the positions of all players are fed into a single GRU network, but independent networks are used to compute latent variables for each agent. This is a mix between VRNN-single and VRNN-indep, which we will call VRNN-mixed, and achieves an ELBO of 2331 and similar statistics as VRNN-indep (we will update Table 1 and Table 3). We’ve also included some generated samples at (https://bit.ly/2S66iO9). However, we emphasize that this model remains fundamentally different from our solution, as our solution provides a degree of controllability and interpretability (through macro-intents) not offered by these baselines.
>
> > “How is the threshold for macro-intent generation selected.”
>
> The threshold is chosen such that it qualitatively matches realistic basketball behavior (i.e. when a basketball player is considered stationary). However, this is a very interesting question raised by the reviewer regarding the effect of labeling functions on the stability and robustness of the model. One can imagine other domains where labeling functions come from a variety of sources, some of which are noisy or redundant. Designing an algorithm that can process these labels and incorporate them into sample generation is a new line of research that we are very excited about.
>
> > “... could using separate [macro-intent] vector for each agent … give the same result?
>
> In the basketball setting, individual macro-intents are in fact sufficient for generating corresponding trajectories. However, this is mainly an architectural detail that is domain-dependent and not the most important part of our contributions. For example, one can also define macro-intents that cannot be factorized for each agent, such as friendly/unfriendly behavior in the Boids model included in our experiments.
>
> > Minor Comments
>
> The results come from sampling from the posterior distribution. The average standard deviation of the learned posterior distribution is around 0.08 per latent dimension. The standard deviation of the learned likelihood of the data is very peaked (often less than 0.01). The macro-intent RNN model achieves a log-likelihood of 2180, which is an improvement over the RNN-gauss model but still worse than all VRNN models. We will update the paper to correct for typos.

---

### Official Review · AnonReviewer1 · 2018-11-10
**Paper proposes multi-agent sequential generative models. This is influential beyond toy simulations presented in the paper.**

**Rating:** 7
**Confidence:** 3

**Review:**

Very strong paper, building on top of variational RNNs for multi-agent sequential generation. Dialogue use case is mentioned in Discussion is indeed very exciting. The approach extends VRNN to a hierarchical setup with high level coordination via a shared learned latent variable. The evaluations are not very strong due to toy task setup, however the approach is clear and impactful.

---

> ### Author Response · Authors · 2018-11-20
> **AnonReviewer1 Response**
>
> Thank you for reviewing our paper and providing insightful feedback. We respond to your main points below.
>
> > “The evaluations are not very strong due to toy setup.”
>
> We emphasize that, although we use a 2D perspective of the game of basketball, this setting of modeling multi-agent tracking data is still highly non-trivial due to the following reasons:
> - Such data is often fine-grained and spans long time horizons.
> - Models must reason over all possible multi-agent trajectories, which is exponentially large w.r.t. the number of agents and time horizon.
> - Expert behavior is often inherently non-deterministic (being unpredictable on offense) and current methods struggle to accurately capture such multimodal behavior.
> - Modeling the coordination between agents is crucial for generating realistic trajectories (e.g. executing a specific offensive play in basketball).
>
> Our approach provides an efficient solution that addresses all the aforementioned challenges, whereas current state-of-the-art baselines perform very poorly in this task (e.g. players going out of bounds, players not moving cohesively, etc.). See (http://bit.ly/2DAu1Ub) for some comparisons, which is the same link provided in the footnote on page 5. Lastly, we comment that coaches and sports analysts evaluate team strategies using a 2D view of the game, so our solution in this space is practically relevant.

---

### Author Response · Authors · 2018-11-20
**General comments to revewiers**

We thank all reviewers for their insightful comments and will make updates to the paper as needed. We briefly summarize our contributions below.

We work in a novel sequential modeling setting in which the target phenomenon (coordinated multi-agent behavior) is inherently non-deterministic and multimodal. Current approaches do not scale to the complexity of this problem because the space of all possible multi-agent trajectories is exponentially large w.r.t. the number of agents, and the agents are often highly coordinated.

We propose an efficient solution that uses a simple labeling function in sequential generative models to learn a macro-intent latent variable that encodes long-term intent and captures the coordination between agents. Our results demonstrate that our model generates trajectories of significantly higher quality than current baselines. Lastly, we highlight that our approach provides a degree of control and interpretability not offered by other baselines; the macro-intent variables are well understood (since they originate from a heuristic labeling function) and their effect on generated samples can be easily analyzed.

We believe that this work opens a new line of research into algorithms that can provide users with various degrees of control during sample generation. Current alternatives involve learning latent variables in a fully unsupervised fashion and inspecting them after training for interpretable features. Our work uses labeling functions to directly control sample generation in ways that can be specified by the user. For example, the labeling function we used for basketball allows users to control where they want players to go (see Figure 6a in our paper). We are very excited about future work in this direction.

---

### Meta-Review · Area_Chair1 · 2018-12-13
**Generative models to produce coordinated multi-agent behavior**

**Confidence:** 3
**Recommendation:** Accept (Poster)

**Metareview:**

The paper presents generative models to produce multi-agent trajectories. The approach of  using a simple heuristic labeling function that labels variables that would otherwise be latent in training data is novel and and results in higher quality than the previously proposed baselines.
In response to reviewer suggestions, authors included further results with models that share parameters across agents as well as agent-specific parameters and further clarifications were made for other main comments (i.e., baselines that train the hierarchical model by maximizing an ELBO on the marginal likelihood?).